# LEARNING TO REACH GOALS WITHOUT REINFORCEMENT LEARNING

## ABSTRACT

Imitation learning algorithms provide a simple and straightforward approach for training control policies via standard supervised learning methods. By maximizing the likelihood of good actions provided by an expert demonstrator, supervised imitation learning can produce effective policies without the algorithmic complexities and optimization challenges of reinforcement learning, at the cost of requiring an expert demonstrator – typically a person – to provide the demonstrations. In this paper, we ask: can we use imitation learning to train effective policies without any expert demonstrations? The key observation that makes this possible is that, in the multi-task setting, trajectories that are generated by a suboptimal policy can still serve as optimal examples for other tasks. In particular, in the setting where the tasks correspond to different goals, every trajectory is a successful demonstration for the state that it actually reaches. Informed by this observation, we propose a very simple algorithm for learning behaviors without any demonstrations, user-provided reward functions, or complex reinforcement learning methods. Our method simply maximizes the likelihood of actions the agent actually took in its own previous rollouts, conditioned on the goal being the state that it actually reached. Although related variants of this approach have been proposed previously in imitation learning settings with example demonstrations, we present the first instance of this approach as a method for learning goal-reaching policies entirely from scratch. We present a theoretical result linking self-supervised imitation learning and reinforcement learning, and empirical results showing that it performs competitively with more complex reinforcement learning methods on a range of challenging goal reaching problems.

## 1 INTRODUCTION

Reinforcement learning (RL) algorithms hold the promise of providing a broadly-applicable tool for automating control, and the combination of high-capacity deep neural network models with RL extends their applicability to settings with complex observations and that require intricate policies. However, RL with function approximation, including deep RL, presents a challenging optimization problem. Despite years of research, current deep RL methods are far from a turnkey solution: most popular methods lack convergence guarantees (Baird, 1995; Tsitsiklis & Van Roy, 1997) or require prohibitive numbers of samples (Schulman et al., 2015; Lillicrap et al., 2015). Moreover, in practice, many commonly used algorithms are extremely sensitive to hyperparameters (Henderson et al., 2018). Besides the optimization challenges, another usability challenge of RL is reward function design: although RL automatically determines *how* to solve the task, the task itself must be specified in a form that the RL algorithm can interpret and optimize. These challenges prompt us to consider whether there might exist a general method for learning behaviors without the need for complex, deep RL algorithms.

Imitation learning is an alternative paradigm to RL that provides a simple and straightforward approach for training control policies via standard supervised learning methods. By maximizing the likelihood of good actions provided by an expert demonstrator, supervised imitation learning can produce effective policies without the algorithmic complexities and optimization challenges of RL. Supervised learning algorithms in deep learning have matured to the point of being robust and reliable, and imitation learning algorithms have demonstrated success in acquiring behaviors robustly and reliably from high-dimensional sensory data such as images (Rajeswaran et al., 2017; Lynch

et al., 2019). The catch is that imitation learning methods require an expert demonstrator – typically a human – to provide a number of demonstrations of optimal behavior. Obtaining expert demonstrations can be challenging; the large number of demonstrations required limits the scalability of such algorithms. In this paper, we ask: can we use ideas from imitation learning to train effective policies without *any* expert demonstrations, retaining the benefits of imitation learning, but making it possible to learn goal-directed behavior autonomously from scratch?

The key observation for making progress on this problem is that, in the multi-task setting, trajectories that are generated by a suboptimal policy can serve as optimal examples for other tasks. In particular, in the setting where the tasks correspond to reaching different goal states, every trajectory is a successful demonstration for the state that it *actually* reaches. Similar observations have been made in prior works as well (Kaelbling, 1993; Andrychowicz et al., 2017; Nair et al., 2018; Mavrin et al., 2019; Savinov et al., 2018), but have been used to motivate data reuse in off-policy RL or semiparametric methods. Our approach will leverage this idea to obtain near-optimal goal-conditioned policies without RL or reward functions.

The algorithm that we study is, at its core, very simple: at each iteration, we run our latest goal-conditioned policy, collect data, and then use this data to train a policy with supervised learning. Supervision is obtained by noting that each action that is taken is a good action for reaching the states that actually occurred in future time steps along the same trajectory. This algorithm resembles imitation learning, but is *self-supervised*. This procedure combines the benefits of goal-conditioned policies with the simplicity of supervised learning, and we theoretically show that this algorithm corresponds to a convergent policy learning procedure. While several prior works have proposed training goal-conditioned policies via imitation learning based on a superficially similar algorithm (Ding et al., 2019; Lynch et al., 2019), to our knowledge no prior work proposes a complete policy learning algorithm based on this idea that learns from scratch, without expert demonstrations. This procedure reaps the benefits of off-policy data re-use without the need for learning complex $Q$ functions or value functions. Moreover, we can bootstrap our algorithm with a small number of expert demonstrations, such that it can continue to improve its behavior self supervised, without dealing with the challenges of combining imitation learning with off-policy RL.

The main contribution of our work is a complete algorithm for learning policies from scratch via goal-conditioned imitation learning, and to show that this algorithm can successfully train goal-conditioned policies. Our theoretical analysis of self-supervised goal-conditioned imitation learning shows that this method optimizes a lower bound on the probability that the agent reaches the desired goal. Empirically, we show that our proposed algorithm is able to learn goal reaching behaviors from scratch without the need for an explicit reward function or expert demonstrations.

## 2 RELATED WORK

Our work addresses the same problem statement as goal conditioned reinforcement learning (RL) (Andrychowicz et al., 2017; Held et al., 2018; Kaelbling, 1993; Nair et al., 2018), where we aim to learn a policy via RL that can reach different goals. Learning goal-conditioned policies is quite challenging, especially when provided only sparse rewards. This challenge can be partially mitigated by hindsight relabeling approaches that relabel goals retroactively (Kaelbling, 1993; Schaul et al., 2015; Pong et al., 2018; Andrychowicz et al., 2017). However, even with relabelling, the goal-conditioned optimization problem still uses unstable off-policy RL methods. In this work, we take a different approach and leverage ideas from supervised learning and data relabeling to build off-policy goal reaching algorithms which do not require any explicit RL. This allows GCSL to inherit the benefits of supervised learning without the pitfalls of off-policy RL. While, in theory, on-policy algorithms might be used to solve goal reaching problem as well, their inefficient use of data makes it challenging to apply these approaches to real-world settings.

Our algorithm is based on ideas from imitation learning (Billard et al., 2008; Hussein et al., 2017) via behavioral cloning (Pomerleau, 1989) but it is not an imitation learning method. While it is *built* on top of ideas from supervised learning, we are not trying to imitate externally provided expert demonstrations. Instead, we build an algorithm which can learn to reach goals from scratch, without explicit rewards. A related line of work (Hester et al., 2018; Brown et al., 2019) has explored how agents can leverage expert demonstrations to bootstrap the process of reinforcement learning. While GCSL is an algorithm to learn goal-reaching policies from scratch, it lends itself naturally

to bootstrapping from demonstrations. As we show in Section 5.4, GCSL can easily incorporate demonstrations into off-policy learning and continue improving, avoiding many of the challenges described in Kumar et al. (2019b).

Recent imitation learning algorithms propose methods that are closely related to GCSL. Lynch et al. (2019) aim to learn general goal conditioned policies from "play" data collected by a human demonstrator, and Ding et al. (2019) perform goal-conditioned imitation learning where expert goal-directed demonstrations are relabeled for imitation learning. However, neither of these methods are iterative, and both require human-provided expert demonstrations. Our method instead iteratively performs goal-conditioned behavioral cloning, starting from scratch. Our analysis shows that performing such iterated imitation learning on the policy's *own* sampled data actually optimizes a lower bound on the probability of successfully reaching goals, without the need for *any* expert demonstrations.

The cross-entropy method (Mannor et al., 2003), self-imitation learning (Oh et al., 2018), reward-weighted regression (Peters & Schaal, 2007), path-integral policy improvement (Theodorou et al., 2010), reward-augmented maximum likelihood (Norouzi et al., 2016; Nachum et al., 2016), and proportional cross-entropy method (Goschin et al., 2013) selectively weight policies or trajectories by their performance during learning, as measured by then environment's reward function. While these may appear procedurally similar to GCSL, our method is fully self-supervised, as it does not require a reward function, and is applicable in the goal-conditioned setting. Additionally, our algorithm continues to perform well in the purely off-policy setting, where no new data is collected, a key difference from other algorithms (Lynch et al., 2019; Ding et al., 2019).

A few works similar to ours in spirit study the problem of learning goal-conditioned policy without external supervision. Zero-shot visual imitation uses an inverse model with forward consistency to learn from novelty seeking behavior, but lacks convergence guarantees and requires learning a complex inverse model Pathak et al. (2018). Semi-parametric methods (Savinov et al., 2018; Eysenbach et al., 2019) learn a policy similar to ours but do so by building a connectivity graph over the visited states in order to navigate environments, which requires large memory storage and computation time that increases with the number of states.

## 3 PRELIMINARIES

**Goal Reaching** We consider the goal reaching in an environment defined by the tuple $\langle \mathcal{S}, \mathcal{A}, \mathcal{T}, \rho(s_0), T, p(g) \rangle$. $\mathcal{S}$ and $\mathcal{A}$ correspond to the state and action spaces respectively, $\mathcal{T}(s'|s,a)$ to the transition kernel, $\rho(s_0)$ to the initial state distribution, $T$ the horizon length, and $p(g)$ to the distribution over goal states $g \in \mathcal{S}$. We aim to find a time-varying goal-conditioned policy $\pi(\cdot|s,g,h)$: $\mathcal{S} \times \mathcal{S} \times [T] \to \Delta(\mathcal{A})$, where $\Delta(\mathcal{A})$ is the probability simplex over the action space $\mathcal{A}$ and $h$ is the remaining horizon. We will say that a policy is optimal if it maximizes the probability the specified goal is reached at the end of the episode:

$$J(\pi) = \mathbb{E}_{g \sim p(g)} \left[ P_{\pi_g}(s_T = g) \right]. \tag{1}$$

It is important to note here that we are not considering the notion of optimality to be finding the *shortest* path to the goal, but merely saying that the trajectory must reach the goal at the end of $T$ time-steps. This problem can equivalently be cast in reinforcement learning. The modified state space $\mathcal{S}' = \mathcal{S} \times \mathcal{S} \times [T]$ contains the current state, goal, and the remaining horizon; the modified transition kernel $\mathcal{T}'((s', g', h') \mid (s, g, h), a) = \mathcal{T}(s' \mid s, a) \cdot \mathbb{1}(d = d', h' = h - 1)$ appropriately handles the modified state space; and the reward function $r((s, g, h)) = \mathbb{1}(s = g, h = 0)$ depends on both the goal and the time step. Because of the special structure of this formulation, off-policy RL methods can relabel an observed transition $((s, g, h), a, (s', g, h - 1))$ to that of a different goal $g'$ and different horizon $h'$ like $((s, g', h'), a, (s', g', h' - 1))$. A common approach is to relabel trajectories with the goal they *actually* reached instead of the *commanded* goal, and often referred to as hindsight experience replay (Andrychowicz et al., 2017; Nair et al., 2018).

**Imitation Learning** We consider algorithms for goal-reaching that use behavior cloning, a standard method for imitation learning. In behavior cloning for goal-conditioned policies, an expert policy provides demonstrations for reaching some target goals at the very last timestep, and we aim to find a policy that best predicts the expert actions from the observations. More formally,

given a dataset $\mathcal{D} = \{\{s_0^0, a_0^0, s_1^0, a_1^0, ....s_T^0\}, \{s_0^1, a_0^1, s_1^1, a_1^1, ....s_T^1\}, ...\}$, and a set of stochastic, time-varying policies $\Pi$, the behavior-cloned policy corresponds to

$$\pi_{BC} = \arg\max_{\pi \in \Pi} \mathbb{E}_{(s_{T-h}, a_{T-h}, s_T) \sim \mathcal{D}} \left[ \log \pi(a_{T-h} | s_{T-h}, s_T, h) \right].$$

Goal-conditioned imitation learning (Lynch et al., 2019) investigate a similar formalism that makes an additional assumption on the quality of the expert demonstrations: that the expert is optimal not just for reaching $s_T$, but also optimal for reaching all the states $s_1, \ldots s_{T-1}$ preceding it. This corresponding policy is

$$\pi_{GCIL} = \arg\max_{\pi \in \Pi} \mathbb{E}_{(s_t, a_t, s_{t+h}) \sim \mathcal{D}} \left[ \log \pi_\theta(a_t | s_t, s_{t+h}, h) \right] \qquad \text{for } t, h > 0 \text{ and } t + h \le T. \quad (2)$$

Note that Lynch et al. (2019) implement a special case of this objective where the policy is independent of the horizon. In the next section, we will discuss how repeatedly alternating between data collection and goal-conditioned imitation learning can be used to learn a goal-reaching policy. Perhaps surprisingly, this procedure optimizes the objective in Equation 1, without relying on expert demonstrations.

## 4 LEARNING GOAL-CONDITIONED POLICIES WITH SELF-IMITATION

The goal-conditioned imitation learning results in prior work show that expert demonstrations can provide supervision not only for the task the expert was aiming for, but also for reaching any state along the expert's trajectory (Lynch et al., 2019; Ding et al., 2019). Can we design a procedure that uses goal-conditioned behavior cloning as a subroutine, that does not need *any* expert demonstrations, but that nonetheless optimizes a well-defined reward function?

In this work, we show how the idea of imitation learning with data relabeling can be re-purposed to construct a learning algorithm that is able to learn how to reach goals from scratch without any expert demonstrations. We shed light on the reasons that imitation learning with data relabeling is so powerful, and how building an iterative algorithm out of this procedure gives rise to a method that optimizes a lower bound on a reinforcement learning objective, while providing a number of benefits over standard RL algorithms. It is important to note here that we are *not* proposing an imitation learning algorithm, but an algorithm for learning how to reach goals from scratch without any expert demonstrations. We are simply leveraging ideas from imitation learning to build such a goal reaching algorithm.

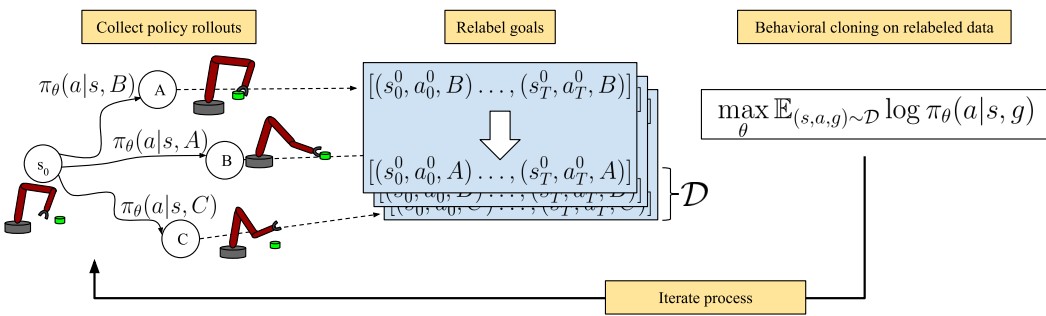

Figure 1: Goal conditioned supervised learning: We can learn how to reach goals by simply sampling trajectories, relabeling them to be optimal in hindsight and treating them as expert data, and then performing supervised learning via behavior cloning.

### 4.1 GOAL REACHING VIA ITERATED IMITATION LEARNING

First, consider goal conditioned imitation learning via behavioral cloning with demonstrations (Equation 2). This scheme works well given expert data $\mathcal{D}$, but expert data is unavailable when we are learning to reach goals from scratch. Can we use goal conditioned behavior cloning to learn how to reach goals from scratch, without the need for *any* expert demonstrations?

To leverage behavior cloning when learning from scratch, we use the following insight: while an arbitrary trajectory from a sub-optimal policy may be suboptimal for reaching the intended goal, it may be optimal for reaching some *other* goal. In the goal-reaching formalism defined in Equation 1, recall a policy is optimal if it maximizes the probability that the goal is reached at the *last* timestep of an episode. This notion of optimality doesn't have to take the direct or shortest path to the goal, it simply has to eventually reach the goal. Under this notion of optimality, we can use a simple data relabeling scheme to construct an expert dataset from an arbitrary set of trajectories. Consider a trajectory $\tau = \{s_1, a_1, s_2, a_2, \ldots, s_T, a_T\}$ obtained by commanding the policy $\pi_\theta(a|s, g, h)$ to reach some goal $g$. Although the actions may be suboptimal for reaching the commanded goal $g$, they do succeed at reaching the states $s_{t+1}, s_{t+2}, \ldots$ that occur later in the observed trajectory. More precisely, for any timestep $t$ and horizon $h$, the action $a_t$ in state $s_t$ is likely to be a good action for reaching $s_{t+h}$ in $h$ timesteps, and thus useful supervision for $\pi_\theta(\cdot|s_t, s_{t+h}, h)$. This step of autonomous relabeling allows us to convert suboptimal trajectories into optimal goal reaching trajectories for different goals, without the need for any human intervention. To obtain a concrete algorithm, we can relabel all such timesteps and horizons in a trajectoryto create an expert dataset according to $\mathcal{D}_\tau = \{(s_t, a_t, s_{t+h}, h) : t, h > 0, t + h \leq T\}$. Because the relabelling procedure is valid for any horizon $h \leq T$, we can relabel every such combination to create $\binom{T}{2}$ optimal tuples of $(s, a, g, h)$ from a single trajectory.

This relabeled dataset can then be used to perform goal-conditioned behavioral cloning to update the policy $\pi_\theta$. While performing one iteration of goal conditioned behavioral cloning on the relabeled dataset is not immediately sufficient to reach all desired goals, we will show that this procedure does in fact optimize a lower bound on a well-defined reinforcement learning objective. As described concretely in Algorithm 1, the algorithm proceeds as follows: (1) Sample a goal from a target goal distribution $p(g)$. (2) Execute the current policy $\pi(a|s, g, h)$ for $T$ steps in the environment to collect a potentially suboptimal trajectory $\tau$. (3) Relabel the trajectory according to the previous paragraph to add $\binom{T}{2}$ new expert tuples $(s_t, a_t, s_{t+h}, h)$ to the training dataset. (4) Perform supervised learning on the entire dataset to update the policy $\pi(a|s, g, h)$ via maximum likelihood. We term this iterative procedure of sampling trajectories, relabelling them, and training a policy until convergence *goal-conditioned supervised learning* (GCSL).

---

**Algorithm 1** Goal-Conditioned Supervised Learning (GCSL)

---

1: **procedure** GCSL
2:      Initialize policy $\pi_1(\cdot \mid s, g, h)$ and dataset $\mathcal{D}((s, a, g, h))$
3:      **for** $k = 1, 2, 3, \ldots$ **do**
4:          Sample $g \sim p(g)$ and execute $\pi_k$ in environment trying to reach $g$
5:          Log trajectory $\tau = (s_0, a_0, s_1, a_1, \ldots s_T, a_T)$
6:          Add tuples $\{(s_t, a_t, s_{t+h}, h) : t, h > 0, t + h \leq T\}$ to the dataset $\mathcal{D}$
7:          Optimize the policy $\pi_{k+1} \leftarrow \arg\max_{\pi_\theta} \mathbb{E}_{(s,a,g,h)\sim\mathcal{D}} \left[\log \pi_\theta(a \mid s, g, h)\right]$
8:      **end for**
9: **end procedure**

---

The GCSL algorithm (described above) provides us with an algorithm that can learn to reach goals from the target distribution $p(g)$ simply using iterated behavioral cloning. The resultant goal reaching algorithm is off-policy, uses low variance gradients, and is simple to implement and tune without the need for any explicit reward function engineering or demonstrations. Additionally, since this algorithm is off-policy and does not require a value function estimator, it is substantially easier to bootstrap from demonstrations when real demonstrations are available, as our experiments will show in Section 5.4.

### 4.2 THEORETICAL JUSTIFICATION

While the GCSL algorithm is simple to implement, does this algorithm actually solve a well-defined policy learning problem? In this section, we argue that GCSL maximizes a lower bound on the probability for a policy to reach commanded goals.

We start by writing the probability that policy $\pi$ conditioned on goal $g$ produces trajectory $\tau$ as $\pi(\tau|g) = p(s_0) \prod_{h=0}^{T} \pi(a_t|s_t, g, h)\mathcal{T}(s_{t+1}|s_t, a_t)$. We define $\mathcal{G}(\tau) = s_T$ as final state of a trajec-

tory. Recalling Equation 1, the target goal-reaching objective we wish to maximize is the probability of reaching a commanded goal:

$$J(\pi) = \mathbb{E}_{g \sim p(g)} \left[ P_{\pi_g}(s_T = g) \right] = \mathbb{E}_{\substack{g \sim p(g), \\ \tau \sim \pi(\tau|g)}} \left[ \mathbb{1}[\mathcal{G}(\tau) = g] \right].$$

In the language of reinforcement learning, we are optimizing a multi-task problem where the reward in each task is an indicator that a goal was reached. The distribution over tasks (goals) of interest is assumed to be pre-specified as $p(g)$. In the on-policy setting, GCSL performs imitation learning on trajectories commanded by the goals that were reached by the current policy, an objective that can be written as

$$J_{\text{GCSL}}(\pi) = \mathbb{E}_{\tau \sim \mathbb{E}_g[\pi_{old}(\cdot|g)]} \left[ \sum_{h=0}^{T} \log \pi(a_t|s_t, \mathcal{G}(\tau), h) \right].$$

Here, $\pi_{old}$ corresponds to a copy of $\pi$ through which gradients do not propagate, following the notation of Schulman et al. (2015). Our main result, which is derived in the on-policy data collection setting, shows that optimizing $J_{\text{GCSL}}(\pi)$ optimizes a lower bound on the desired objective, $J(\pi)$ (proof in Appendix B):

**Theorem 4.1.** *Let $J_{GCSL}$ and $J$ be as defined above. Then, $J(\pi) \geq J_{GCSL}(\pi) + C$, Where C is a constant that does not depend on $\pi$.*

Note that to prevent $J(\pi)$ and $J_{GCSL}(\pi)$ from being zero, the probability of reaching a goal under $\pi$ must be nonzero - in scenarios where such a condition does not hold, the bound remains true, albeit vacuous. The tightness of this bound can be controlled by the effective error in the GCSL objective - we present this, alongside a technical analysis of the bound in Appendix $B.1$. This indicates that in the regime with expressive policies where the loss function can be minimized well, GCSL will improve the expected reward.

## 5 EXPERIMENTS

In our experimental evaluation, we aim to answer the following questions:

1. Does GCSL effectively learn goal-conditioned policies from scratch?

2. Does the performance of GCSL improve over successive iterations?

3. Can GCSL learn goal-conditioned policies from high-dimensional image observations?

4. Can GCSL incorporate demonstration data more effectively than standard RL algorithms?

### 5.1 EXPERIMENTAL DETAILS

We consider a number of simulated control environments: 2-D room navigation, object pushing with a robotic arm, and the classic Lunar Lander game, shown in Figure 2. The tasks allow us to study the performance of our method under a variety of system dynamics, both low-dimensional state inputs and high-dimensional image observations, and in settings with both easy and difficult exploration. For each task, the target goal distribution corresponds to a uniform distribution over all reachable configurations. Performance of a method is quantified by the distance of the agent to the goal at the last timestep (not by the number of time-steps to the goal as is sometimes considered in goal reaching). We present full details about the environments, evaluation protocol, and hyperparameter choices in Appendix A.

For the practical implementation of GCSL, we parametrize the policy as a neural network that takes in state, goal, and horizon as input and outputs a parametrized action distribution. We find that omitting the horizon from the input to the policy still provides good results, despite the formulation suggesting that the optimal policy is most likely non-Markovian. We speculate that this is due to optimal actions changing only mildly with different horizons in our tasks. Full details about the implementation for GCSL are presented in Appendix A.1.

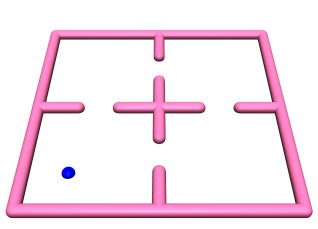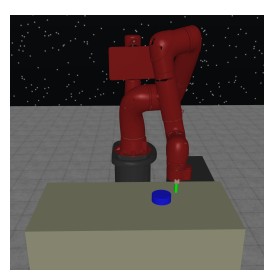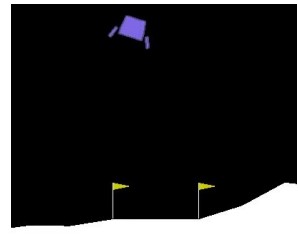

Figure 2: **Evaluation Tasks**: For each of the following tasks, we evaluate our algorithm using low-dimensional sensory data and pixel observations: *(Left)* 2D Navigation, *(Center)* robotic pushing, and *(Right)* Lunar Lander.

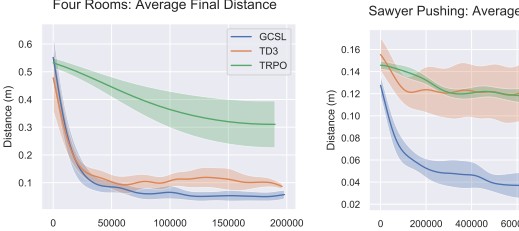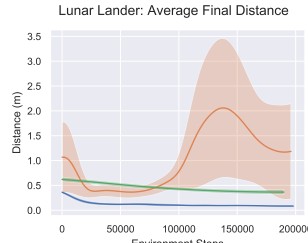

Figure 3: **State-based tasks:** GCSL is competitive with state-of-the-art off-policy value function RL algorithms for goal-reaching from low-dimensional sensory observations. Shaded regions denote the standard deviation across 3 random seeds (lower is better).

## 5.2 LEARNING GOAL-CONDITIONED POLICIES

We evaluate the effectiveness of GCSL for reaching goals on the domains visualized in Figure 2, both from low-dimensional proprioception and from images. To better understand the performance of our algorithm, we compare to two families of reinforcement learning algorithms for solving goal-conditioned tasks. First, we consider off-policy temporal-difference RL algorithms, particular TD3-HER (Eysenbach et al., 2019; Held et al., 2018), which uses hindsight experience replay to more efficiently learn goal-conditioned value functions. TD3-HER requires significantly more machinery than our simple procedure: it maintains a policy, a value function, a target policy, and a target value function, all which are required to prevent degradation of the learning procedure. We also compare with on-policy reinforcement learning algorithms such as TRPO (Schulman et al., 2015) that cannot leverage data relabeling, but often provide more stable optimization than off-policy methods. Because these methods cannot relabel data, we provide an epsilon-ball reward corresponding to reaching the goal. Details for the training procedure for these comparisons, along with hyperpa-

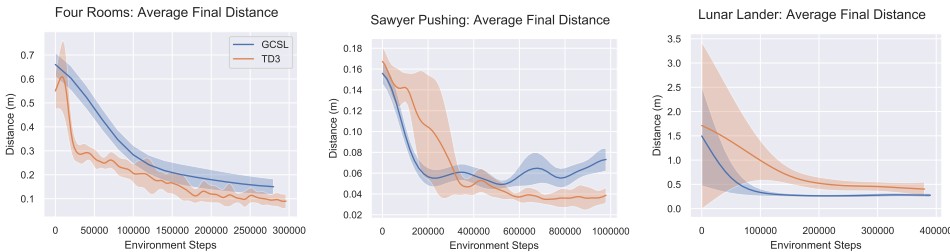

Figure 4: **Image-based tasks**: On three tasks with image observations, GSCL achieves similar performance to a state-of-the-art baseline, TD3, while being substantially simpler. Shaded regions denote the standard deviation across 3 random seeds (lower is better).

rameter and architectural choices, are presented in Appendix A.2. Videos and further details can be found at `https://sites.google.com/view/gcsl/`.

We first investigate the learning performance of these algorithms from low-dimensional sensor observations, as shown in Figure 3. We find that on the pushing and lunar lander domains, GCSL is able to reach a larger set of goals consistently than either RL algorithm. Although TD3 is able to fully solve the navigation task, on the other domains which require synthesis of more challenging control behavior, the algorithm makes slow, if any, learning progress. Given a limited amount of data, TRPO performs poorly as it cannot relabel or reuse data, and so cannot match the performance of the other two algorithms. When scaling these algorithms to image-based domains, which we evaluate in Figure 4, we find that GCSL is still able to learn goal-reaching behaviors on several of these tasks. albeit slower than from state. For most tasks, from both state and images, GCSL is able to reach within 80% of the desired goals and learn at a rate comparable to or better than previously proposed off-policy RL methods. This evaluation demonstrates that simple iterative self-imitation is a competitive scheme for reaching goals in challenging environments which scales favorably with dimensionality of state and complexity of behaviors.

## 5.3 ANALYSIS OF LEARNING PROGRESS AND LEARNED BEHAVIORS

To better understand the learning behaviors of the algorithm, we investigate how GCSL performs as we vary the quality and quantity of data, the policy class we optimize over, and the relabelling technique (Figure 5). Full details for these scenarios can be found in Appendix A.4.

First, we consider how varying the policy class can affect the performance of GCSL. In Section 5.1, we hypothesized that optimizing over a Markovian policy class would be performant over maintaining a non-Markovian policy. We find that allowing policies to be time-varying ("Time-Varying Policy" in Figure 5) can drastically speed up training on small domains, as these non-Markovian optimal policies can be fit more closely. However, on domains with active exploration challenges such as the Pusher, exploration using time-varying policies is ineffective, and degrades performance.

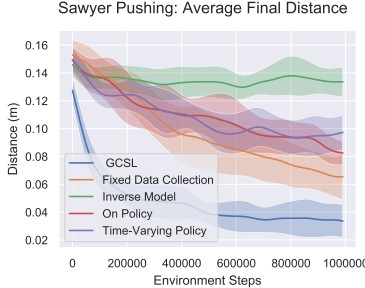

Figure 5: Performance across variations of GCSL (Section 5.3) for the pushing domain. Plots for other domains in Appendix A.4

We investigate how the quality of the data in the dataset used to train the policy affects the learned policy. We consider two variations of GCSL: one which collects data using a fixed policy ("Fixed Data Collection" in Figure 5) and another which limits the size of the dataset to be small, forcing all the data to be on-policy ("On-Policy" in Figure 5). When collecting data using a fixed policy, the learning progress of the algorithm demonstratedly decreases, which indicates that the iterative loop of collecting data and training the policy is crucial for converging to a performant solution. By forcing the data to be all on-policy, the algorithm cannot utilize the full set of experiences seen thus far and must discard data. Although this on-policy process remains effective on simple domains, the technique leads to slower learning progress on tasks requiring more challenging control.

## 5.4 INITIALIZING WITH DEMONSTRATIONS

Because GCSL can perform self-imitation from arbitrary data sources, the algorithm is amenable to initialization from prior exploration or from demonstrations. In this section, we study how GCSL performs when incorporating expert demonstrations as initializations. Our results comparing GCSL and TD3 in this setting corroborate the existing hypothesis that off-policy value function RL algorithms are challenging to integrate with initialization from demonstrations Kumar et al. (2019a).

We consider the setting where an expert provides a set of demonstration trajectories, each for reaching a different goal. GCSL requires no modifications to incorporate these demonstrations - it simply adds the expert data to the initial dataset, and begins the training procedure. In contrast, multiple prior works have proposed *additional* algorithmic changes to off-policy TD-based methods to in-

corporate data from expert demonstrations (Kumar et al., 2019a). We compare the performance of GCSL to one such variant of TD3-HER in incorporating expert demonstrations on the robotic pushing environment in Figure 6 (Details in Appendix A.5). Although TD3 achieves better performance with the demonstrations than when learning from scratch, it drops in performance at the beginning of training, which means TD3 regresses from the initial behavior-cloned policy, an undesirable characteristic for initializing from demonstrations. In contrast, GCSL scales favorably, learns faster than from scratch, and effectively incorporates the expert demonstrations. We believe this benefit largely comes from not needing to train an explicit critic, which can be unstable when trained using highly off-policy data such as demonstrations (Kumar et al., 2019b).

## 6 DISCUSSION AND FUTURE WORK

In this work, we proposed GCSL, a simple algorithm for learning goal-conditioned policies that uses imitation learning, while still learning autonomously from scratch. This method is exceptionally simple, relying entirely on supervised learning to learn policies by relabeling its own previously collected data. This method can easily utilize off-policy data, seamlessly incorporate expert demonstrations when they are available, and can learn directly from image observations. Although several prior works have explored similar algorithm designs in an imitation learning setting (Ding et al., 2019; Lynch et al., 2019), to our knowledge our work is the first to derive a complete iterated algorithm based on this principle for learning from scratch, and the first to theoretically show that this method optimizes a lower bound on a well-defined reinforcement learning objective.

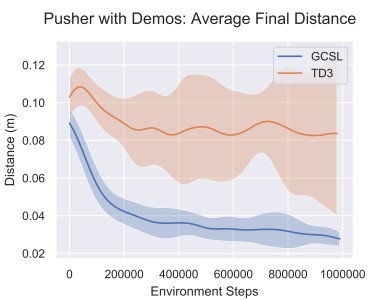

Figure 6: Initializing from Demonstrations: GCSL is more amenable to initializing using expert demonstrations than value-function RL methods.

While our proposed method is simple, scalable, and readily applicable, it does have a number of limitations. The current instantiation of this approach provides limited facilities for effective exploration, relying entirely on random noise during the rollouts to explore. More sophisticated exploration methods, such as exploration bonuses (Mohamed & Rezende, 2015; Storck et al., 1995), are difficult to apply to our method, since there is no explicit reward function that is used during learning. However, a promising direction for future work would be to reweight the sampled rollouts based on novelty to effectively incorporate a novelty-seeking exploration procedure. A further direction for future work is to study whether the simplicity and scalability of our method can make it possible to perform goal-conditioned reinforcement learning on substantially larger and more varied datasets. This can in principle enable wider generalization, and realize a central goal in goal-conditioned reinforcement learning — universal policies that can succeed at a wide range of tasks in diverse environments.

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

# A    EXPERIMENTAL DETAILS

## A.1    GOAL-CONDITIONED SUPERVISED LEARNING (GCSL)

GCSL iteratively performs maximum likelihood estimation using a dataset of relabelled trajectories that have been previously collected by the agent. Here we present details about the policy class, data collection procedure, and other design choices. We parametrize a time-invariant policy using a neural network which takes as input state and goal, and returns probabilities for a discretized grid of actions of the action space. For the state-based domains, the neural network is a feedforward network with two hidden layers of size $400$ and $300$ respectively. For the image-based domains, both the observation image and the goal image are first preprocessed through three convolutional layers, with kernel size $5, 3, 3$ and channels $16, 32, 32$ respectively. When executing in the environment, data is sampled according to an exploratory policy which increases the temperature of the current policy: $\pi_{explore}(a|s, g) \propto \pi(a|s, g)^\alpha$. The replay buffer stores trajectories and relabels on the fly, with the size of the buffer subject only to memory constraints.

## A.2    RL COMPARISONS

We perform experimental comparisons with TD3-HER (Fujimoto et al., 2018; Andrychowicz et al., 2017). We relabel transitions as $((s, g), a, (s', g))$ gets relabelled to $((s, g'), a, (s', g'))$, where $g' = g$ with probability $0.1$, $g' = s'$ with probability $0.5$, and $g' = s_t$ for some future state in the trajectory $s_t$ with probability $0.4$. As described in Section 3, the agent receives a reward of $1$ and the trajectory ends if the transition is relabelled to $g' = s'$, and $0$ otherwise. Under this formalism, the optimal $Q$-function, $Q * (s, a, g) = \exp(-T(s, g))$, where $T(s, g)$ is the minimum expected time to go from $s$ to $g$. Both the Q-function and the actor for TD3 are parametrized as neural networks, with the same architecture (except final layers) for state-based domains and image domains as those for GCSL.

We also compare to TRPO (Schulman et al., 2015), an on-policy RL algorithm. Because TRPO is on-policy, we cannot relabel goals, and so we provide a surrogate $\epsilon$-ball indicator reward function: $r(s, g) = 1(d(s, g) < \epsilon)$, where $\epsilon$ is chosen appropriately for each environment. To maximize the data efficiency of TRPO, we performed a coarse hyperparameter sweep over the batch size for the algorithm. Just as with TD3, we mimic the same neural network architecture for the parametrizations of the policies as GCSL.

## A.3    TASK DESCRIPTIONS

For each environment, the goal space is identical to the state space. For the image-based experiments, images were rendered at resolution $84 \times 84 \times 3$.

**2D Room Navigation** This environment requires an agent to navigate to points in an environment with four rooms that connect to adjacent rooms. The state space has two dimensions, consisting of the cartesian coordinates of the agent. The agent has acceleration control, and the action space has two dimensions. The distribution of goals $p(g)$ is uniform on the state space, and the agent starts in a fixed location in the bottom left room.

**Robotic Pushing** This environment requires a Sawyer manipulator to move a freely moving block in an enclosed play area with dimensions $40$ cm $\times$ $20$ cm. The state space is 4-dimensional, consistsing of the cartesian coordinates of the end-effector of the sawyer agent and the cartesian coordinates of the block. The Sawyer is controlled via end-effector position control with a three-dimensional action space. The distribution of goals $p(g)$ is uniform on the state space (uniform block location and uniform end-effector location), and the agent starts with the block and end-effector both in the bottom-left corner of the play area.

**Lunar Lander** This environment requires a rocket to land in a specified region. The state space includes the normalized position of the rocket, the angle of the rocket, whether the legs of the rocket are touching the ground, and velocity information. Goals are sampled uniformly along the landing region, either touching the ground or hovering slightly above, with zero velocity.

### A.4 ABLATIONS

In Section 5.3, we analyzed the performance of the following variants of GCSL (Figure 7).

1. **Inverse Model** - This model relabels only states and goals that are one step apart: $\{(s_t, a_t, s_{t+h}, h) : t > 0, h = 1\}$

2. **On-Policy** Only the most recent 10000 transitions are stored and trained on.

3. **Fixed Data Collection** Data is collected according to a uniform policy over actions.

4. **Time-Varying Policy** Policies are are conditioned on the remaining horizon. Alongside the state and goal, the policy gets a reverse temperature encoding of the remaining horizon as input.

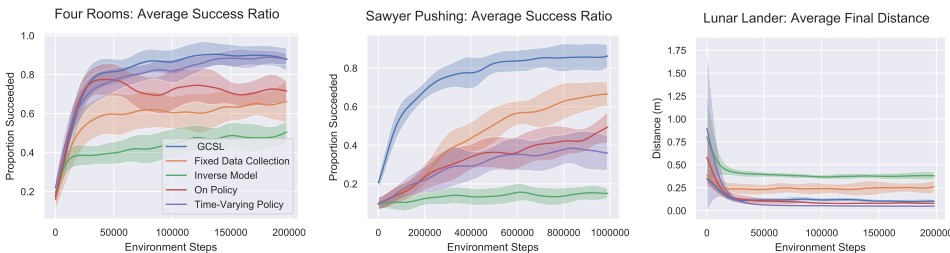

Figure 7: Performance across variations of GCSL (Section 5.3) for all three experimental domains.

### A.5 INITIALIZING WITH DEMONSTRATIONS

We train an expert policy for robotic pushing using TRPO with a shaped dense reward function, and collect a dataset of 200 trajectories, each corresponding to a different goal. To train GCSL using these demonstrations, we simply populate the replay buffer with these trajectories at the beginning of training, and optimize the GCSL objective using these trajectories to warm-start the algorithm. Initializing a value function method using demonstrates requires significantly more attention: we perform the following procedure. First, we perform goal-conditioned behavior cloning to learn an initial policy $\pi_{BC}$. Next, we collect 200 new trajectories in the environment using a uniform data collection scheme. Using this dataset of 400 trajectories, we perform policy evaluation on $\pi_{BC}$ to learn $Q^{\pi_{BC}}$ using policy evaluation via bootstrapping. Having trained such an estimate of the Q-function, we initialize the policy and Q-function to these estimates, and run the appropriate value function RL algorithm.

## B PROOF OF THEOREM 4.1

We will assume a discrete state space in this proof, and denote a trajectory as $\tau = \{s_0, a_0, \ldots, s_T, a_T\}$. Let the notation $\mathcal{G}(\tau) = s_T$ denote the final state of a trajectory, which represents the goal that the trajectory reached. As there can be multiple paths to a goal, we let $\tau_g = \{\tau : \mathcal{G}(\tau) = g\}$ denote the set of trajectories that reach a particular goal $g$. We abbreviate a policy's trajectory distribution as $\pi(\tau|g) = p(s_0) \prod_{t=0}^{T} \pi(a_t|s_t, g)\mathcal{T}(s_{t+1}|s_t, a_t)$. The target goal-reaching objective we wish to optimize is the probability of reaching a commanded goal,

$$J(\pi) = \mathbb{E}_{g \sim p(g), \tau \sim \pi(\tau|g)}[\mathbb{1}[\mathcal{G}(\tau) = g]]$$

The distribution over tasks (goals) is assumed to be pre-specified as $p(g)$. GCSL optimizes the following objective, where the log-likelihood of the actions conditioned on the goals actually reached by the policy, $\mathcal{G}(\tau)$:

$$J_{\text{GCSL}}(\pi) = \mathbb{E}_{\tau \sim \mathbb{E}_g[\pi_{old}(\cdot|g)]}\left[\sum_{t=0}^{T} \log \pi(a_t|s_t, \mathcal{G}(\tau))\right]$$

Here, using notation from Schulman et al. (2015), $\pi_{old}$ is a copy of the policy $\pi$ through which gradients do not propagate. To analyze how this objective relates to $J(\pi)$, we first analyze the relationship between $J(\pi)$ and a *surrogate objective*, given by

$$J_{\mathrm{surr}}(\pi) = E_{g \sim p(g)} \left[ \sum_{\tau \in \tau_g} \pi_{old}(\tau|g) \log \pi(\tau|g) \right]$$

As $J_{surr}(\pi)$ and $J(\pi)$ have the same gradient for all $\pi$, the differ by some $\pi$-independent constant $C_1$, i.e. $J(\pi) = J_{surr}(\pi) + C_1$.

We can now lower-bound the surrogate objective via the following:

$$
\begin{aligned}
J_{\mathrm{surr}}(\pi) &= E_{g \sim p(g)} \left[ \sum_{\tau \in \tau_g} \pi_{old}(\tau|g) \log \pi(\tau|g) \right] \\
&= E_{g \sim p(g)} \left[ \sum_{\tau} \mathbb{1}[\mathcal{G}(\tau) = g] \pi_{old}(\tau|g) \log \pi(\tau|\mathcal{G}(\tau)) \right] \\
&= \sum_g p(g) \sum_\tau \log \pi(\tau|\mathcal{G}(\tau)) \pi_{old}(\tau|g) \mathbb{1}[\mathcal{G}(\tau) = g] \\
&= \sum_\tau \log \pi(\tau|\mathcal{G}(\tau)) \sum_g p(g) \pi_{old}(\tau|g) \mathbb{1}[\mathcal{G}(\tau) = g] \\
&= \sum_\tau \log \pi(\tau|\mathcal{G}(\tau)) \sum_g p(g) \pi_{old}(\tau|g) - \sum_\tau \log \pi(\tau|\mathcal{G}(\tau)) \sum_g p(g) \pi_{old}(\tau|g) \mathbb{1}[\mathcal{G}(\tau) \neq g] \\
&\qquad\qquad (3) \\
&\geq \sum_\tau \log \pi(\tau|\mathcal{G}(\tau)) \sum_g p(g) \pi_{old}(\tau|g) \\
&= \mathbb{E}_{\tau \sim \mathbb{E}_g[\pi_{old}(\tau|g)]}[\log \pi(\tau|\mathcal{G}(\tau))].
\end{aligned}
$$

The final line is our goal-relabeling objective: we train the policy to reach goals we reached $g'$. The inequality holds since $\log \pi(\tau)$ is always negative. The inequality is loose by a term related to the probability of not reaching the commanded goal.

Since the initial state and transition probabilities do not depend on the policy, we can simplify $\log \pi(\tau|\mathcal{G}(\tau))$ as (by absorbing non $\pi$-dependent terms into $C_2$):

$$
\begin{aligned}
\mathbb{E}_{\tau \sim \mathbb{E}_g[\pi_{old}(\tau|g)]}[\log \pi(\tau|\mathcal{G}(\tau))] &= \mathbb{E}_{\tau \sim \mathbb{E}_g[\pi_{old}(\tau|g)]} \left[ \log p(s_0) + \sum_{t=0}^{T} \log \pi(a_t|s_t, \mathcal{G}(\tau)) + \log \mathcal{T}(s_{t+1}|s_t, a_t) \right] \\
&= \mathbb{E}_{\tau \sim \mathbb{E}_g[\pi_{old}(\tau|g)]} \left[ \sum_{t=0}^{T} \log \pi(a_t|s_t \mathcal{G}(\tau)) \right] + C_2 \\
&= J_{\mathrm{GCSL}}(\pi) + C_2.
\end{aligned}
$$

Combining this result with the bound on the expected return completes the proof, namely that $J(\pi) \geq J_{GCSL}(\pi) + C_1 + C_2$. Note that in order for $J(\pi)$ and $J_{GCSL}(\pi)$ to not be degenerate, the probability of reaching a goal under $\pi_{old}$ must be non-zero. This assumption is reasonable, and matches the assumptions on "exploratory data-collection" and full-support policies that are required by Q-learning and policy gradient convergence guarantees.

### B.1 QUANTIFYING THE QUALITY OF THE APPROXIMATION

We now seek to better understand the gap introduced by Equation 3 in the analysis above. We define $P_{\pi_{old}}(\mathcal{G}(\tau) \neq g)$ to be the probability of failure under $\pi_{old}$ and $p(g)$. We overload the notation $\pi_{old}(\tau) = \mathbb{E}_g[\pi_{old}(\tau|g)]$, and additionally define $p_{\mathrm{wrong}}(\tau)$ and $p_{\mathrm{right}}(\tau)$ the conditional distribution of trajectories under $\pi_{old}$ given that it did not reach and did the commanded goal respectively.

In the following section, we show that the gap can be controlled by the probability of making a mistake, $P_{\pi_{old}}(\mathcal{G}(\tau) \neq g)$, and $D_{TV}(p_{\text{wrong}}(\tau), p_{\text{right}}(\tau))$, a measure of the difference between the distribution of trajectories that must be relabelled and those not.

We rewrite Equation 3 as follows:

$$J_{surr}(\pi) = \sum_\tau \log \pi(\tau|\mathcal{G}(\tau)) \sum_g p(g)\pi_{old}(\tau|g) - \sum_\tau \log \pi(\tau|\mathcal{G}(\tau)) \sum_g p(g)\pi_{old}(\tau|g)\mathbb{1}[\mathcal{G}(\tau) \neq g]$$

$$= \mathbb{E}_{\tau \sim \pi_{old}}[\log \pi(\tau|\mathcal{G}(\tau))] - P_{\pi_{old}}(\mathcal{G}(\tau) \neq g))\mathbb{E}_{\tau \sim p_{\text{wrong}}(\tau)}[\log \pi(\tau|\mathcal{G}(\tau))]$$

Define $D$ to be the Radon-Nikodym derivative of $p_{\text{wrong}}(\tau)$ wrt $\pi_{old}(\tau)$

$$= \mathbb{E}_{\tau \sim \pi_{old}}[\log \pi(\tau|\mathcal{G}(\tau))] - P_{\pi_{old}}(\mathcal{G}(\tau) \neq g))\mathbb{E}_{\tau \sim \pi_{old}}[D \log \pi(\tau|\mathcal{G}(\tau))]$$

$$= (1 - P_{\pi_{old}}(\mathcal{G}(\tau) \neq g))\mathbb{E}_{\tau \sim \pi_{old}}[\log \pi(\tau|\mathcal{G}(\tau))]$$
$$+ \underbrace{P_{\pi_{old}}(\mathcal{G}(\tau) \neq g))\mathbb{E}_{\tau \sim \pi_{old}}[(1 - D)\log \pi(\tau|\mathcal{G}(\tau))]}_{\text{Relevant Gap}}$$

The first term is affine with respect to the GCSL loss, so the second term is the error we seek to understand.

$$|\text{Relevant Gap}| = P_{\pi_{old}}(\mathcal{G}(\tau) \neq g)|\mathbb{E}_{\tau \sim \pi_{old}}[(1 - D)\log \pi(\tau|\mathcal{G}(\tau))]|$$
$$\leq P_{\pi_{old}}(\mathcal{G}(\tau) \neq g)\mathbb{E}_{\tau \sim \pi_{old}}[|1 - D|]\mathbb{E}_{\tau \sim \pi_{old}}[\log \pi(\tau|\mathcal{G}(\tau))]$$
$$= 2P_{\pi_{old}}(\mathcal{G}(\tau) \neq g)D_{TV}(\mathbb{E}_g[\pi_{old}(\tau|g)], p_{\text{wrong}}(\tau))\mathbb{E}_{\tau \sim \pi_{old}}[\log \pi(\tau|\mathcal{G}(\tau))]$$
$$= 2P_{\pi_{old}}(\mathcal{G}(\tau) \neq g)(1 - P_{\pi_{old}}(\mathcal{G}(\tau) \neq g))D_{TV}(p_{\text{right}}(\tau), p_{\text{wrong}}(\tau))\mathbb{E}_{\tau \sim \pi_{old}}[\log \pi(\tau|\mathcal{G}(\tau))]$$

The inequality is maintained because of the nonpositivity of $\log \pi(\tau)$, and the final step holds because $\pi_{old}(\tau)$ is a mixture of $p_{\text{wrong}}(\tau)$ and $p_{\text{right}}(\tau)$. This derivation shows that the gap between $J_{surr}$ and $J_{GCSL}$ (up to affine consideration) can be controlled by 1) the probability of reaching the wrong goal and 2) the divergence between the conditional distribution of good trajectories and those which must be relabelled. As either term goes to 0, the bound becomes tight.

We now show that sufficiently optimizing the GCSL objective causes the probability of reaching the wrong goal to be bounded close to 0, and thus bounds the gap close to 0.

Suppose we collect trajectories from a policy $\pi_{data}$. Mimicking the notation from before, we define $\pi_{data}(\tau) = \mathbb{E}_{g \sim p(g)}[\pi_{data}(\tau|g)]$. For convenience, we define $\pi_{data}^*(a_t|s_t, g) \propto \int_{\tau \backslash a_t} \pi_{data}(\tau)1(\mathcal{G}(\tau) = g)1(s_t(\tau) = s_t)$ to be the conditional distribution of actions for a given state given that the goal $g$ is reached at the end of the trajectory. If this conditional distribution is not defined, we let $\pi_{data}^*(a_t|s_t, g)$ be uniform, so that $\pi_{data}^*(a_t|s_t, g)$ is well-defined for all states, goals, and timesteps.

**Lemma B.1.** *Consider an environment with deterministic dynamics in which all goals are reachable in $T$ timesteps, and a behavior policy $\pi_{data}$ which is exploratory: $\pi_{data}(a_t|s_t, g) > 0$ for all $(a, s, g)$ (for example epsilon-greedy exploration). Suppose the GCSL objective is sufficiently optimized, so that for all $s, g \in \mathcal{S}$, and time-steps $t \in \{1, \ldots, T\}$, $D_{TV}(\pi(a_t|s_t, g), \pi_{data}^*(a_t|s_t, g)) \leq \epsilon$.*

*Then, the probability of making a mistake $P_\pi(\mathcal{G}(\tau) \neq g)$ can be bounded above by $\epsilon T$*

*Proof.* We show the result through a coupling argument, similar to Schulman et al. (2015); Kakade & Langford (2002); Ross et al. (2011). Because $D_{TV}(\pi(a_t|s_t, g), \pi_{data}^*(a_t|s_t, g)) \leq \epsilon$, we can define a $(1 - \epsilon)$-coupled policy pair $(\overline{\pi}, \overline{\pi_{data}^*})$, which takes differing actions with probability $\epsilon$. By a union bound over all timesteps, the probability that $\overline{\pi}$ and $\overline{\pi_{data}^*}$ take any different actions throughout the trajectory is bounded by $\epsilon T$, and under assumptions of deterministic dynamics, take the same trajectory with probability $1 - \epsilon T$. Under assumption of deterministic dynamics and all goals being reachable from the initial state distribution in $T$ timesteps, the policy $\pi_{data}^*(a_t|s_t, g)$ satisfies $P_{\pi_{data}^*}(\mathcal{G}(\tau) \neq g) = 0$. Because $\pi_{data}^*$ reaches the goal with probability 1, this implies that $\pi$ must also reach the goal with probability at least $1 - \epsilon T$. Thus, $P_\pi(\mathcal{G}(\tau) \neq g) \leq \epsilon T$. □

## C    EXAMPLE TRAJECTORIES

Figure 8 below shows parts of the state along trajectories produced by GCSL. In Lunar Lander, this state is captured by the rocket's position, and in 2D Room Navigation it is the agent's position. While these trajectories do not always take the shortest path to the goal, they do often take fairly direct paths to the goal from the initial position avoiding very roundabout trajectories.

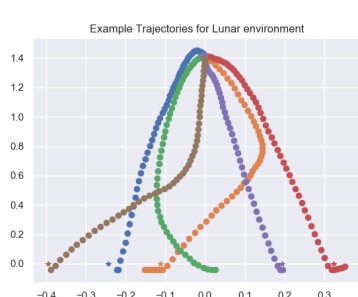 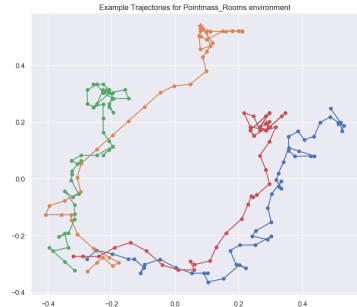

Figure 8: Examples of trajectories generated by GCSL for the Lunar Lander and 2D Room environments. Stars indicate the goal state.

