# OpenReview forum: "Learning to Reach Goals Without Reinforcement Learning"
_ICLR.cc/2020/Conference — Reject_

### Official Review · AnonReviewer1 · 2019-10-22
**Official Blind Review #1**

**Rating:** 6

**Review:**

This work presents the goal-conditioned supervised learning algorithm (GCSL), which learns goal conditioned policies using only behavioral-cloning of the agent's own actions.  The intuition behind the algorithm is the goal of an observed trajectory can be identified after the fact, by simply looking at the states reached during that trajectory.  GCSL treats each executed action as a sample from the expert policy conditioned on each of the states reached after that action is taken.  Given a distribution over goal states, GCSL alternates between executing its current goal-conditioned policy on randomly selected goals, and learning to imitate the generated actions conditioned on the states they actually reached.  Experimental results demonstrate superior performance against a base (non-goal conditioned) RL algorithm (TRPO), and against another approach to learning goal-conditioned polices (TD3-HER), on a relatively diverse set of control problems.

A major issue is that the proof of the main theoretical result appears to be wrong.  As there don't appear to be any constraints placed on the policy pi_old, it would seem that the surrogate loss would collapse to 0 for any policy pi if pi_old is such that the target goal is never reached (the probability of any trajectory t reaching g is 0 under pi_old(t|g)).  It seems to be the case that the quality of the GCSL loss depends on the relationship between pi_old and the goal distribution p(g).  The fact that the theoretical results are incorrect does not mean that the algorithm, or the general approach do not have value, but it does highlight the fact that this approach may only be effective for a specific class of problems similar to the experimental domains.

While not a flaw in the work itself, it should be made clear in the text that the notion of optimality for the learning tasks considered in this work (i.e. achieving the goal by the end of episode), avoids one of the apparent limitations of the algorithm.  A randomly generated trajectory is itself optimal for any state that it reaches, if we define optimality as simply reaching a state.  Such a trajectory may not be the most efficient way of reaching that state however, so the relabelling process would seem to be prone to learning policies that achieve the conditioned goals, but not doing so in an efficient manner.  It isn't clear how well this approach would work for tasks where the efficiency, in terms of the time required to reach the objective, is a key part of the evaluation.  Again, this is not a flaw in the work itself, and it is possible that the algorithm will be effective in such tasks, perhaps because the likelihood of an action resulting in a given state is higher if that action brings us closer to this state.  It might be useful to conduct some additional experiments where evaluation is based on the time required to solve a task, rather than just the accuracy of the final state.

**Experience Assessment:**

I have read many papers in this area.

**Review Assessment: Checking Correctness Of Derivations And Theory:**

I carefully checked the derivations and theory.

**Review Assessment: Checking Correctness Of Experiments:**

I assessed the sensibility of the experiments.

**Review Assessment: Thoroughness In Paper Reading:**

I read the paper at least twice and used my best judgement in assessing the paper.

---

> ### Author Response · Authors · 2019-11-13
> **Response to Reviewer 1**
>
> Thank you for your insightful comments and suggestions! We have revised the paper to address the concern about the "notion of optimality," and we have provided additional theoretical analysis on the relationship between the GCSL loss and J(pi) to address the concerns raised in the review. We provide detailed responses to many of your comments below:
>
> “While not a flaw in the work itself, it should be made clear in the text that the notion of optimality for the learning tasks considered in this work (i.e. achieving the goal by the end of episode), avoids one of the apparent limitations of the algorithm.”
> -> We agree strongly with this point! We have made this discussion more clear in Section 3, 4.1 and 5.1. As you point out, “optimality” for our method means reaching the goal within a fixed time horizon, not reaching it as quickly as possible. To understand the nature of the behaviors learned by GCSL, we have provided a visualization of learned trajectories in Appendix B.1. We find that these behaviors, while not necessarily being shortest path in terms of time-steps to the goal, doesn’t take extremely long paths to the goal either. Please let us know if this addresses your concern, or if you would like to see further revisions to address this point.
>
> “As there don't appear to be any constraints placed on the policy pi_old, ...”
> -> To prevent the surrogate loss from being 0 for a given goal, it is indeed required that the probability of reaching the goal is nonzero for pi_old - we have updated the discussion in Appendix B to clarify this point. Note that this assumption is not unreasonable, and would be required to guarantee convergence of Q-learning or policy gradient approaches as well.
>
> “It seems to be the case that the quality of the GCSL loss depends on the relationship between pi_old and the goal distribution p(g).”
> -> We agree that our proof presents a bound that is overly loose if the desired goal distribution and the experienced state distribution are very different - the given result is not incorrect, but arguably vacuous in such scenarios. We have included a new section in Appendix B which quantifies the gap between the two losses, as a function of the probability of failure and the distribution shift between “relabelled” and “unrelabelled” trajectories. We present a new bound in Lemma B.1 that shows if the GCSL loss is well optimized throughout the state space, the gap between these two losses nears zero. Please let us know if this addresses your concern, or if you would like to see further revisions to address the theory.

---

> > ### Comment · AnonReviewer1 · 2019-11-14
> > **Assumptions on pi_old**
> >
> > Thank you for taking the time to address these concerns.
> >
> > Regarding the assumptions on pi_old, those should be made explicit in Section 4 where the theorem is actually stated.  Even with this assumption though, I am still not certain that the surrogate loss is equivalent to the true loss up to a constant factor.  Specificaly, the gradient of the surrogate loss for a specific goal has trajectories weighted by pi_old, conditioned on the fact that they reach the goal, while the gradient of the true loss has them weighted by pi, again conditioned on the fact that the trajectory reaches the goal.  These distributions might be very different.
> >
> > Regarding the new bound in appendix B, would this bound imply that a form of policy iteration (using exact integration over trajectories) would converge given an initial policy satisfying the assumptions required for Theorem 4.1.

---

> > > ### Author Response · Authors · 2019-11-14
> > > **Clarifications on pi_old and bounds**
> > >
> > > “Regarding the new bound in appendix B, would this bound imply that a form of policy iteration (using exact integration over trajectories) would converge given an initial policy satisfying the assumptions required for Theorem 4.1”
> > > -> The new bound provided by Lemma B.1 implies that given an exploratory data collection policy and a fully expressive (e.g. tabular) policy class, in the limit of infinite data or exact integration over trajectories), we converge to an optimal policy which maximizes the probability of reaching goals in the environment. Note that this kind of infinite sample analysis is typical for such proofs -- e.g., the Trust Region Policy Optimization proofs also only consider the infinite sample limit. Accounting for sampling error in such analysis is generally quite difficult.
> > >
> > > “Regarding the assumptions on pi_old …”
> > > -> We'd like to clarify the statement of Theorem 4.1, which may have been potentially misleading in the original version of the paper - we have updated the paper to clarify this ambiguity. Theorem 4.1 demonstrates that J_{GCSL}(pi) is a lower bound on J(pi) when *on-policy* trajectories from pi are relabelled and trained on for the GCSL objective. Following the notation of Schulman et al 2015a, pi_{old} is not an arbitrary distribution, but rather a copy of the policy pi through which gradients do not propagate. This is the same pi_{old} that appears in surrogate objectives for the REINFORCE policy gradient, and in derivations for Schulman et al 2015b. As also discussed in those works, with this definition of pi_old, the two objectives J(pi) and J_{surr}(pi) may have different values, but have the same gradient for all pi, and thus equivalent to a constant factor. We have updated both Section 4 and Appendix B to clarify the definition of pi_{old} used in Theorem 4.1.
> > >
> > > Please note that although Theorem 4.1 requires on-policy data, the new bound in Lemma B.1 provides performance guarantees that do not depend on on-policy data collection.
> > >
> > > 1. Schulman, J., Levine, S., Moritz, P., Jordan, M., & Abbeel, P.  (2015a). Trust Region Policy Optimization. ICML.
> > > 2. Schulman, J., Heess, N.M., Weber, T., & Abbeel, P. (2015b). Gradient Estimation Using Stochastic Computation Graphs. NIPS.

---

> > > > ### Comment · AnonReviewer1 · 2019-11-15
> > > > **Bounds**
> > > >
> > > > So I believe the on-policy equvalence you describe in the rebuttal is correct when J_GCSL(pi) is evaluated for trajectories sampled from pi (and becomes a weaker approximation as pi and pi_old deviate).  The way Theorem 4.1 is presented just does not make this clear.  I would suggest reorganizing that section to remove incorporate the new bounds from B.1, and state explicitly (perhaps as a Corollary) that the equivalence holds for pi = pi_old.

---

### Official Review · AnonReviewer3 · 2019-10-22
**Official Blind Review #3**

**Rating:** 3

**Review:**

The paper claims to do imitation learning without expert demonstration using trajectories that are generated by suboptimal policies from other tasks.

The point of having an expert demonstrator is to help narrow the search for an optimal policy.  By taking the expert demonstration knowledge out of learning, to me, this is not retaining the benefit of imitation learning.  Thus, the paper is not about imitation learning, but rather about an optimization method that reuses data generated from multiple tasks.   Reusing trajectory data generated from multiple tasks to learn a policy of another task is not a novel idea.  If we could save all of the data regardless of whether if an optimal policy generates them or not, why not use them?  Less useful data may still contain useful information.  The better question is how to use them to learn policy efficiently.  If the motivation is to use trajectories from suboptimal policies from other tasks without expert knowledge, then I fail to see the motivation and the novelty of this paper.

The paper claims that the methodology self-supervises each action taken, judging how good it is for reaching a goal in the future without learning Q-values.  However, this was not realized.  The methodology gathers all trajectories that reach a goal into a set, and use behaviour cloning on the data of the set to learn a policy.  The sampled trajectories in the set could be suboptimal for reaching a goal, and there’s little evidence that optimizing J_GCSL(\pi) will learn an optimal policy based on these data.  Optimizing objective J_GCSL(\pi) also does not take the long term effect of actions into account.  The gathering of trajectories and identifying the trajectory as goal-reaching is already a costly step, where no learning happens.  RL, on the other hand, would gather the data incrementally, learn, and act right away.  Also, it seems that the algorithm would require human knowledge to discern a trajectory as goal-reaching or not, which is contrary to self-supervision.

**Experience Assessment:**

I do not know much about this area.

**Review Assessment: Checking Correctness Of Derivations And Theory:**

I assessed the sensibility of the derivations and theory.

**Review Assessment: Checking Correctness Of Experiments:**

I assessed the sensibility of the experiments.

**Review Assessment: Thoroughness In Paper Reading:**

I read the paper at least twice and used my best judgement in assessing the paper.

---

> ### Author Response · Authors · 2019-11-13
> **Response to Reviewer 3**
>
> Thank you for your comments and feedback! We would like to clarify a few aspects about the GCSL algorithm. We have modified the main text of the paper in Section 4 to make these sections more clear and explicit and to address your concerns. Please let us know if these clarifications address your concerns!
>
> GCSL is *not* an imitation learning algorithm, but rather an algorithm which leverages ideas from imitation learning to learn goal-reaching behaviors from scratch without the need for any expert demonstration trajectories. Our insight is that even though a trajectory may be suboptimal for the goal being attempted to reach, it is optimal to reach the final state of the trajectory. This insight enables us to generate examples of optimal trajectories from potentially suboptimal ones via automated hindsight relabelling. By combining automated hindsight relabelling with the optimization techniques from imitation learning, we are able to devise a goal-reaching algorithm which avoids the need for bootstrapping or complicated policy gradient schemes that are prevalent in current RL algorithms, and can learn without the need for any human demonstrations as well.
>
> “If we could save all of the data regardless of whether if an optimal policy generates them or not, why not use them?  Less useful data may still contain useful information.  The better question is how to use them to learn policy efficiently.”
> -> We absolutely agree about the importance of re-using arbitrary past data to learn policies efficiently! Prior approaches which use all previous data learn policies and value functions via bootstrapping, which is known to be very unstable and difficult to optimize (Kumar et al 2019). What we propose in our paper is a more stable and performant policy optimization scheme borrowing ideas from imitation learning, which is also able to efficiently use all previously collected data, regardless of suboptimality. By performing the automatic hindsight relabelling scheme described in Section 4.1 on all previously collected trajectories, we can transform the policy learning problem into a supervised learning (behavior cloning) objective. This allows us to do policy learning from scratch, while retaining the optimization benefits of supervised learning and imitation learning such as simplicity, stability, scalability to larger neural networks, and easy bootstrapping from demonstrations.
>
> “Also, it seems that the algorithm would require human knowledge to discern a trajectory as goal-reaching or not, which is contrary to self-supervision.”
> -> Since we are using automated relabeling to make use of *all* trajectory data that was collected, there is no need for human knowledge to discern goal-reaching trajectories. Could you clarify what you mean by human knowledge in this case? We believe this may stem from a misreading of the paper, which we are eager to correct.
>
> “The sampled trajectories in the set could be suboptimal for reaching a goal, and there’s little evidence that optimizing J_GCSL(\pi) will learn an optimal policy based on these data.”
> -> While a trajectory may be suboptimal for reaching the goal that it was trying to reach, after the relabeling step (described in Section 4.1 and line 6 of Algorithm 1),  the trajectory becomes optimal for the relabeled goal under the notion of optimality defined in Equation 1. This is important because this can now be treated as expert data to optimize J_{GCSL} correctly.
>
> “The gathering of trajectories and identifying the trajectory as goal-reaching is already a costly step, where no learning happens.  RL, on the other hand, would gather the data incrementally, learn, and act right away”
> -> GCSL actually incurs the same data collection complexity as more traditional RL algorithms. Prior works developing RL algorithms for goal-reaching [Eysenbach et al, Lin et al] also perform trajectory gathering and relabelling prior to training the policy and value function. Although we presented GCSL in separate data collection, relabelling, and training substeps, all three of these processes can be performed concurrently, just as you mentioned.
>
>
> 1. Eysenbach, B., Salakhutdinov, R., & Levine, S. (2019). Search on the Replay Buffer: Bridging Planning and Reinforcement Learning.
> 2. Lin, X., Baweja, H.S., & Held, D. (2019). Reinforcement Learning without Ground-Truth State. ArXiv, abs/1905.07866.

---

> > ### Comment · AnonReviewer3 · 2019-11-13
> > **Elaborate on how relabeling is performed**
> >
> > The relabeling of data has been mentioned in the paper, but how is it actually performed?  In particular, how is an action a_t in s_t measured to be a good action for reaching distant s_{t+h} such that the tuple (s_t, a_t, s_{t+h}, h) is added to the relabeled dataset?  The practical implication of this approach depends on the details of this relabeling step.

---

> > > ### Author Response · Authors · 2019-11-14
> > > **Clarification of relabeling scheme**
> > >
> > > For a trajectory {s_0, a_0, s_1, a_1, .... s_T}, we relabel every such tuple (s_t, a_t, s_{t+h}, h) to the dataset (a total of O(T^2) tuples for one trajectory). This may seem counterintuitive, but this relabelling strategy arises as a consequence of the particular notion of optimality we seek to maximize  (defined in Section 3) - the likelihood of reaching the goal within a time limit of T timesteps. Under this notion of optimality, an optimal trajectory need not find the shortest path to the goal, but rather simply must reach the goal at the desired time-limit. If we witness some trajectory containing the snippet (s_t, a_t, s_{t+1}, a_{t+1}, .... s_{t+h}), this confirms the existence of a path from s_t to s_{t+h} which takes h timesteps when taking action a_t. Therefore, a_t at s_t must be optimal to reach s_{t+h} for a policy which is attempting to reach the goal exactly *h* timesteps in the future. where The lack of restrictions on when this relabelling can be done allows us to reuse data aggressively. Although in theory this may lead to "lazy" trajectories which wait or initially go the wrong way, we find in practice that the policy learns generally straightforward paths to the goal (as visualized in Appendix C). We have updated Section 4 of the paper to clarify our relabelling scheme. Please let us know if this addresses your concerns about which tuples are relabelled.

---

### Official Review · AnonReviewer2 · 2019-10-24
**Official Blind Review #2**

**Rating:** 6

**Review:**

This paper proposes a method to learn to reach goals in an RL environment. The method is based on principles of imitation learning. For instance, beginning with an arbitrary policy that samples a sequence of state-action pairs, in the next iteration, the algorithm treats the previous policy as an expert by relabeling its ending state as a goal. The paper shows that the method is theoretically sound and effective empirically for goal-achieving tasks.

The paper is relatively clear and experiments are okay. I would then recommend it is on the positive side of the borderline.

Comments:
* The method is interesting but is still an "RL" method. So it is really learning to reach the goal via "RL". Note that in the method, the algorithm is not doing effective exploration but just randomly explore until you collect sufficient data to solve for a new goal.
* If you formulate the problem better, you can see that it actually has a reward: add an initial state s0; for each g sampled from p(g), transition s0 to an MDP with goal g. You can now do the usual RL algorithm in this new MDP. I would think you can also do model-based learning -- give the model a good representation and then use the policies to learn the dynamics. It may worth to compare your algorithm with these natural baselines.


**Experience Assessment:**

I have published in this field for several years.

**Review Assessment: Checking Correctness Of Derivations And Theory:**

I assessed the sensibility of the derivations and theory.

**Review Assessment: Checking Correctness Of Experiments:**

I assessed the sensibility of the experiments.

**Review Assessment: Thoroughness In Paper Reading:**

I made a quick assessment of this paper.

---

> ### Author Response · Authors · 2019-11-13
> **Response to Reviewer 2**
>
> Thank you for your insightful comments and suggestions! We have updated the paper to address your concerns about the connection between our method and RL, and the role of exploration in GCSL. Please find detailed responses to your comments below:
>
> “The method is interesting but is still an "RL" method.”
>
> -> We agree that goal-reaching can be written as an RL problem! In Paragraph 2 of Section 3 (Preliminaries), we describe an explicit equivalence between our formulation and RL with a sparse indicator reward for reaching the goal. Due to this equivalence, our method implicitly maximizes the reward function defined in Section 3, and thus is an “RL method”. However, unlike more standard “RL methods” like TD3 or TRPO (which we compare to in our experiments), we do not rely on dynamic programming or complex policy gradient schemes, but simply use supervised learning as a subroutine in acquiring goal reaching behaviors. We have updated Section 3 to more clearly describe the MDP formulation for goal-reaching and the connections between our algorithm and other RL methods.
>
> “Note that in the method, the algorithm is not doing effective exploration but just randomly explore until you collect sufficient data to solve for a new goal.”
>
> -> We agree that exploration for our method can be improved! With our current exploration strategy (adding action noise), the quality of exploration is influenced greatly by performance - as the agent becomes better at reaching the goals it has seen, the probability of reaching goals on the fringe that have not been encountered previously increases. That said, most RL methods utilize exploration strategies similar to GCSL -- e.g,  TRPO and PPO use Gaussian policies, DDPG and HER add time-correlated noise, etc. While dedicated exploration methods such as pseudocounts, intrinsic motivation, and RND could improve exploration, we believe this is an orthogonal direction to the current contribution.
>
> “If you formulate the problem better, you can see that it actually has a reward”
> -> We agree that our algorithm is implicitly optimizing an indicator reward, and for that reason we include two baselines which compare with using the same reward as our method and running model-free RL via TD3 or TRPO. We find that these algorithms perform comparably or worse to GCSL despite being much more complex. We are not certain about what you were suggesting with the model-based baselines, would you be able to provide us more details about your suggestion so we can run the comparison.

---

### Decision · Program_Chairs · 2019-12-19

**Decision:**

Reject

**Comment:**

The authors present an algorithm that utilizes ideas from imitation learning to improve on goal-conditioned policy learning methods that rely on RL, such as hindsight experience replay.  Several issues of clarity and the correctness of the main theoretical result were addressed during the rebuttal period in way that satisfied the reviewers with respect to their concerns in these areas.  However, after discussion, the reviewers still felt that there were some fundamental issues with the paper, namely that the applicability of this method to more general RL problems (complex reward functions rather than signle state goals, time ) is unclear.  The basic idea seems interesting, but it needs further development, and non-trivial modifications, to be broadly applicable as an approach to problems that RL is typically used on.  Thus, I recommend rejection of the paper at this time.